# KV-CoRE: Benchmarking Data-Dependent Low-Rank Compressibility of KV-Caches in LLMs

## Abstract

Large language models rely on kv-caches to avoid redundant computation during autoregressive decoding, but as context length grows, reading and writing the cache can quickly saturate GPU memory bandwidth. Recent work has explored KV-cache compression, yet most approaches neglect the data-dependent nature of kv-caches and their variation across layers. We introduce **KV-CoRE** (**KV**-cache **Co**mpressibility by **R**ank **E**valuation), an SVD-based method for quantifying the data-dependent low-rank compressibility of kv-caches. KV-CoRE computes the optimal low-rank approximation under the Frobenius norm and, being gradient-free and incremental, enables efficient dataset-level, layer-wise evaluation. Using this method, we analyze multiple models and datasets spanning five English domains and sixteen languages, uncovering systematic patterns that link compressibility to model architecture, training data, and language coverage. As part of this analysis, we employ the Normalized Effective Rank as a metric of compressibility and show that it correlates strongly with performance degradation under compression. Our study establishes a principled evaluation framework and the first large-scale benchmark of kv-cache compressibility in LLMs, offering insights for dynamic, data-aware compression and data-centric model development.

## 1 Introduction

Large language models (LLMs) adopt the Transformer architecture (Vaswani et al., 2017) and generate text autoregressively under a causal mask, which ensures that past key and value vectors can be cached (KV-cache) (Ott et al., 2019). These caches are stored in high-bandwidth memory (HBM) on GPUs and repeatedly fetched into compute-unit registers during decoding, reducing computation but introducing a memory-bandwidth bottleneck as context length grows. This challenge has motivated both hardware innovation (Rhee et al., 2025) and algorithmic approaches to KV-cache compression (Shi et al., 2024), with our work focusing on the latter.

A natural way to reduce KV-cache cost is to compress key and value representations into lower-dimensional spaces. Many methods use low-rank approximation of projection matrices (Ji et al., 2025; Chang et al., 2024), but they ignore the data-dependent nature of key/value activations, whose intrinsic rank can be smaller in domain-specific tasks that exercise only part of the model's capacity (Yu & Wu, 2023). Moreover, most approaches apply the same compression ratio across layers (Wang et al., 2024), overlooking distinct compressibility profiles. Methods for analyzing and comparing key/value rank across layers remain underexplored.

To address these limitations, we propose KV-CoRE (KV-cache Compressibility by Rank Evaluation), an incremental singular value decomposition (SVD) method that directly operates on key and value features computed over large datasets. Unlike approaches that approximate projection weights, KV-CoRE is data-dependent and captures the intrinsic rank of the KV-cache induced by real inputs. It supports independent per-layer decomposition without cross-layer coupling, is gradient-free, and enables batch-wise computation with low memory overhead. KV-CoRE guarantees a globally optimal low-rank projection under the Frobenius norm, and unlike existing methods (Chen et al., 2021b; Wang et al., 2024) that only compute the optimal compression matrix, it explicitly decomposes long sequences of key and value features to recover the singular value distributions of each layer on a given dataset. This allows systematic evaluation of compressibility and provides an effective diagnostic tool for understanding representational capacity usage in LLMs.

We conduct extensive experiments across open-source LLMs of different sizes and architectures, spanning datasets in instruction following, code generation, medical QA, and multilingual tasks. We introduce normalized effective rank (NER) as a lightweight metric for per-layer compressibility and systematically compare key and value ranks across models and domains. Beyond static evaluation, end-to-end SVD-based truncation shows that NER correlates strongly with perplexity, validating it as a reliable proxy for compression sensitivity. These results reveal consistent layer-wise and data-dependent patterns in KV-cache compressibility, laying the groundwork for dynamic and adaptive strategies.

Our contributions are as follows:

- We propose KV-CoRE, an SVD-based, data-dependent framework for analyzing KV-cache compressibility, which enables efficient dataset-level, layer-wise evaluation with provably optimal low-rank approximations.

- We introduce NER as a metric of compressibility and demonstrate its strong correlation with perplexity and GPT-score, validating it as a reliable tool for evaluating and comparing models.

- We conduct extensive experiments across models, domains, and languages, uncovering systematic patterns that link compressibility to model architecture, training data, and language coverage.

## 2 KV-CoRE METHOD

To evaluate and compress KV caches, we develop an efficient method that incrementally computes the SVD of keys and values over large datasets, enabling layer-wise and data-dependent evaluation of their compressibility in LLMs. At the same time, our method computes the optimal compression matrices, addressing challenges identified in prior work Chen et al. (2021b); Wang et al. (2024); Yuan et al. (2024).

To illustrate our method and its advantages, we first introduce notations, building upon the preliminaries provided in Appendix A.1. As our method applies uniformly across layers and to both key and value spaces, we omit layer-specific notation in the rest of the section and focus on $K$ as an example in the following discussion for simplicity. Consider a dataset containing $l$ tokens, for a particular LLM, let $X = [\mathbf{x}_1; ...; \mathbf{x}_l] \in \mathbb{R}^{l \times d_e}$ be the sequence of activations for an attention layer, computed based on the dataset. Let $W^K$ be the key projection weights of a layer. The corresponding key $K = [\mathbf{k}_1; ...; \mathbf{k}_l]$ features are computed as $K = [\mathbf{k}_1; ...; \mathbf{k}_l] = XW^K$. The data-dependent optimal low-rank approximation problem, introduced in Chen et al. (2021b); Wang et al. (2024); Yuan et al. (2024), is formulated as follows. For each key projection matrix $W^K$, we seek a low-rank matrix $\widetilde{W}^K$ with rank $k$ that minimizes the compression error,

$$\arg\min_{\widetilde{W}^K} ||XW^K - X\widetilde{W}^K||_F^2 \quad \text{s.t.} \quad \text{rank}(\widetilde{W}^K) = k \tag{1}$$

The solution $\widetilde{W}^K$ can be expressed as a pair of down- and up-projection matrices, which compress key vectors into dimension $k$ and then reconstruct them for efficient autoregressive inference.

### 2.1 SVD-BASED KV-CACHE ANALYSIS

Our key idea is to perform SVD on key and value spaces for each layer, allowing an analytical study of the layer-wise compressibility of KV caches given a LLM and a particular dataset. For example, the SVD on $K \in \mathbb{R}^{l \times m_h d_h}$ can be written as $K = \mathcal{U}\Sigma\mathcal{V}^T$, where $\mathcal{U} \in \mathbb{R}^{l \times l}$ and $\mathcal{V} \in \mathbb{R}^{m_h d_h \times m_h d_h}$ are left and right singular values, respectively, and $\Sigma \in \mathbb{R}^{l \times m_h d_h}$ denotes singular values. By the Eckart-Young-Mirsky theorem Eckart & Young (1936), the truncation of the largest singular values along with corresponding singular vectors $K_k = \mathcal{U}_k \Sigma_k \mathcal{V}_k^T$ forms the best $k$-rank approximation to $K$ in the Frobenius norm sense, with the minimal approximation error calculated as follows,

$$||K - K_k||_F^2 = \begin{cases} \sigma_{k+1}, & \text{for the 2-norm} \\ (\sum_{j=k+1}^r \sigma_j)^{\frac{1}{2}}, & \text{for the Frobenious norm} \end{cases} \tag{2}$$

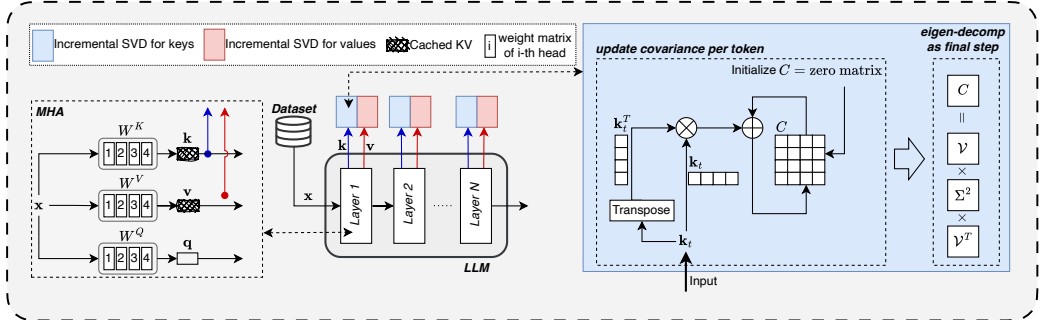

Figure 1: **Overview of KV-CoRE**. At each Transformer layer, keys and values from all attention heads are concatenated to form the KV-cache. To measure compressibility, we apply incremental SVD to the cached key/value activations with low memory overhead, recovering their singular value spectra. The resulting singular vectors and values are used to compute the optimal data-dependent compression matrix, enabling KV-cache compressibility analysis via NER.

where $r$ denotes the rank of $K$, $\sigma_j$ denotes the $j$-th largest singular value in $\Sigma$. In other words, for any $k \in [1, r]$, the minimal $k-$rank approximation loss to $K$ is a deterministic function of singular values. Thus, we use singular value–based metrics to evaluate the compressibility of key and value spaces in each attention layer, as detailed in the experimental section 2.4.

## 2.2 Optimal Data-dependent Compression Matrix

Given singular values $\Sigma$ and right singular vectors $\mathcal{V}$ of $K$, we can directly recover the dataset-dependent optimal $k$-rank approximation of $W^K$ as: $\widetilde{W}^K = W^K \mathcal{V}_k \mathcal{V}_k^T$.

To see why $\widetilde{W}^K$ is optimal to minimize the error: $||X^{(i)}W^{K(i)} - X^{(i)}\widetilde{W}^{K(i)}||_F$, consider the following:

$$
\begin{aligned}
X\widetilde{W}^K = XW^K \mathcal{V}_k \mathcal{V}_k^T &\overset{T_1}{=} K\mathcal{V}_k \mathcal{V}_k^T \\
&\overset{T_2}{=} \mathcal{U}\Sigma(\mathcal{V}^T \mathcal{V}_k)\mathcal{V}_k^T \\
&\overset{T_3}{=} \mathcal{U}\left(\Sigma \begin{bmatrix} I_k \\ 0 \end{bmatrix}\right)\mathcal{V}_k^T = \left(\mathcal{U}\begin{bmatrix} \Sigma_k \\ 0 \end{bmatrix}\right)\mathcal{V}_k^T = \mathcal{U}_k \Sigma_k \mathcal{V}_k^T
\end{aligned}
\tag{3}
$$

Where $I_k$ is a $k$ by $k$ identity matrix. $T_1$ holds because by definition $K = XW^K$; $T_2$ holds because $\mathcal{U}\Sigma\mathcal{V}^T$ is the SVD of $K$; $T_3$ holds because singular vectors form an orthogonal basis, thus $\mathcal{V}_k^T \mathcal{V}_k$ yields a $k$ by $k$ identify matrix. Recall that by the Eckart-Young-Mirsky theorem Eckart & Young (1936), $\mathcal{U}_k \Sigma_k \mathcal{V}_k^T$ forms the best $k$-rank approximation of $K = XW^K$, we can conclude that $W^K \mathcal{V}_k \mathcal{V}_k^T$ is optimal to the optimization problem.

During LLM inference, a direct implementation is to replace $W^K$ with a pair of down-projection $W^K \mathcal{V}_k$ and up-projection $\mathcal{V}_k^T$ matrices. This allows caching the low-dimensional key vector $\mathbf{x}_t W^K \mathcal{V}_k$ instead of the full-dimensional $\mathbf{x}_t W^K$ given t-th token, reducing both the memory and bandwidth consumption.

## 2.3 Incremental SVD Algorithm for Dataset-level KV-cache

A key challenge is that the size of $K \in \mathbb{R}^{l \times m_h d_h}$ scales up with the number of tokens $l$ in a Dataset, rendering direct SVD on $K$ impractical due to excessive memory and computational demands.

We propose a novel algorithm that mitigates memory and computation bottlenecks through batch computation without sacrificing accuracy. This method only requires holding and updating a $m_h d_h$-by-$m_h d_h$ covariance matrix, avoiding the direct SVD of $K$ and still generating the mathematically

equivalent singular values $\Sigma$ and right singular vectors $\mathcal{V}$. Algorithm 1 shows the pseudo-code of our method.

---

**Algorithm 1** SVD Computation of Dataset-level KV-Cache

---

    **Input:** Dataset containing $l$ tokens in total; LLM model weights $W^K$
    **Output:** singular values $\Sigma$ and right singular vectors $\mathcal{V}$ of $K$
  $C \leftarrow m_h d_h \times m_h d_h$ zero matrix                         ▷ Initialize covariance matrix to zero
  **for** $t = 1, \ldots, l$ **do**
      $\mathbf{k}_t \leftarrow \mathbf{x}_t W^K$
      $C \leftarrow C + \mathbf{k}_t^T \mathbf{k}_t$                    ▷ Update covariance matrix in each step
  **end for**
  $\mathcal{V}, \Sigma^2, \mathcal{V}^T \leftarrow$ eigen-decomposition($C$)     ▷ perform eigen-decomposition on the final covariance matrix
  **return** $\Sigma, \mathcal{V}$

---

For each token in a Dataset containing $l$ tokens, $\mathbf{k}_t^T \mathbf{k}_t$ is computed and the covariance matrix $C$ is updated. After $l$ iterations, we will have the complete covariance matrix of $K$ as $C = K^T K = \sum_{t=1}^l \mathbf{k}_t^T \mathbf{k}_t$. Finally we perform eigen-decomposition on $C$ to obtain the singular values $\Sigma$ and right singular vectors $\mathcal{V}$ of $K$.

To see why the final step of Algorithm 1 generates mathematically equivalent $\Sigma$ and $\mathcal{V}$, consider the following proof,

$$K^T K \overset{T_1}{=} (\mathcal{U}\Sigma\mathcal{V}^T)^T \mathcal{U}\Sigma\mathcal{V}^T = \mathcal{V}\Sigma^T \mathcal{U}^T \mathcal{U}\Sigma\mathcal{V}^T \overset{T_2}{=} \mathcal{V}\Sigma^2\mathcal{V}^T \tag{4}$$

where $T_1$ holds by applying SVD on $K$; $T_2$ holds because singular vectors form an orthogonal basis, thus $\mathcal{U}^T \mathcal{U}$ yields a identity matrix, and consequently $\Sigma^T \mathcal{U}^T \mathcal{U}\Sigma = \Sigma^2$.

### 2.4 NORMALIZE EFFECTIVE RANK AS COMPRESSIBILITY METRIC

Intuitively, the approximation loss derived in Eq.(2) provides a direct measure of the compressibility of key and value spaces. For instance, one could fix an approximation rank $k$ across all key spaces, evaluate the corresponding losses, and interpret smaller losses as indicative of higher compressibility. While being simple and straightforward, such strategy has two main drawbacks: First, the metric depends on a fixed approximation rank $k$, and there is no clear guidance on how to select $k$ appropriately. Second, it fails to account for the full spectrum of singular values. We thus introduce the Normalized Effective Rank (NER) as a metric to measure the compressibility of key and value spaces. For a matrix $K$ with singular values $\{\sigma_i\}$ and rank $r$, the effective rank—first introduced in (Roy & Vetterli, 2007)—is defined as $\text{erank}(K) = \exp(-\sum_{i=1}^r p_i \log p_i)$, where $p_i = \sigma_i / \sum_{j=1}^r \sigma_j$, and the logarithm is to the base $e$ (natural logarithm). Building on this, NER is defined as $\text{NER}(K) = \text{erank}(K)/r$. In other words, NER normalizes the effective rank by the matrix's actual rank. As proven in (Roy & Vetterli, 2007), the effective rank $\text{erank}(K)$ satisfies $1 \leq \text{erank}(K) \leq r$, consequently, NER yields a score in $[1/r, 1]$.

## 3 RELATED WORK

**Rank Analysis in Language Models:** Early work has investigated the relationship between the rank of transformer weights or representations and model performance, seeking either to leverage low-rank structure for efficiency (Chen et al., 2021a; Hsu et al., 2022; Hajimolahoseini et al., 2022; Li et al., 2023), to prevent rank collapse that limits expressivity (Dong et al., 2021; Noci et al., 2022; Yaras et al., 2024), or to maximize rank utilization for enhanced modeling capacity (Bhojanapalli et al., 2020; Boix-Adsera et al., 2023). With the growing use of large language models (LLMs), research has turned to their inherent low-rank properties. LoRA (Hu et al., 2022) leverages this structure during fine-tuning, showing that many weight updates lie in low-dimensional subspaces. Loki (Singhania et al., 2024) examined the key representations in attention layers and found that they often reside in lower-dimensional subspaces across models and datasets, which can be used for efficient sparse attention. These directions have also motivated growing efforts on KV-cache

compression (Shi et al., 2024) to address the deployment bottleneck in reading and storing the KV cache (Yu et al., 2022). Our work introduces a fine-grained method for evaluating the compressibility of KV caches in LLMs through effective rank analysis, uncovering layer-wise and data-dependent patterns that can inform the design of dynamic and adaptive compression strategies.

**Low-Rank KV-cache Compression:** DeepSeek (Liu et al., 2024) introduced Multi-head Latent Attention (MLA), which applies low-rank joint KV cache compression to enable scalable inference, unlike Multi-Query Attention (MQA) (Shazeer, 2019) and Grouped-Query Attention (GQA) (Ainslie et al., 2023) which reduce KV caches by merging key and value heads in multi-head attention (MHA) (Vaswani et al., 2017). MHA2MLA (Ji et al., 2025) and PALU (Chang et al., 2024) applies SVD to compress key and value projection weights, converting models based on MHA into the MLA structure for reduced KV cache size. However, this approach targets only the projection weights, while prior work (Yu & Wu, 2023) has shown that transformer weights typically have higher rank than the output features (keys/values), suggesting that data-dependent KV-cache compression is more effective. In this direction, DRONE (Chen et al., 2021b) proposed a closed-form solution for data-aware low-rank compression of projected keys/values, and SVD-LLM (Wang et al., 2024) introduced an incremental optimization based on Cholesky decomposition (Meyer, 2023) that achieves the same optimal compression loss with lower memory overhead. In comparison, our method achieves the same optimality with a much simpler formulation; moreover, it explicitly computes the SVD of key/value representations, whereas SVD-LLM only recovers the optimal compression matrix.

## 4 EXPERIMENT

We evaluate our method on 5 open-source LLM series of varying sizes on 5 datasets.

**Models**    To evaluate the generality of our method across different architectures and model sizes, we apply our analysis to a set of open-source LLMs including Qwen3 (4B, 8B) (Team, 2025), Mistral-7B (Jiang et al., 2023), Gemma-1.1 (2B, 7B) (Team et al., 2024), and Phi-3-mini-128k-instruct (Abdin et al., 2024), where Gemma-1.1[1] is a recent update of the original instruction-tuned Gemma, incorporating a new RLHF method that improves overall performance.

**Datasets**    To study the data-dependence of KV-cache compression, we evaluate our method on $n$ datasets, spanning diverse English instruction-following tasks across multiple domains and multilingual QA. For English evaluation, the datasets cover general instruction following, code generation, medical QA, and function calling, including Alpaca (Taori et al., 2023), MedAlpaca (Han et al., 2023), CodeAlpaca (Chaudhary, 2023), WizardCoder (Luo et al., 2025), and FunctionCall[2]. For multilingual evaluation, we use the queries from the multilingual split of VisR-Bench (Chen et al., 2025), a question-driven, retrieval benchmark spanning 15 languages (Spanish, Italian, German, French, Dutch, Arabic, Croatian, Japanese, Swedish, Vietnamese, Portuguese, Finnish, Czech, Slovenian, and Danish)—allowing us to assess performance across a linguistically diverse setting.

**Hardware and Software Setup**    All experiments are conducted on machines equipped with 8× NVIDIA A800 GPUs (80GB each), though all evaluations are executed on a single GPU without distributed computation. We use PyTorch 2.7.1 and Hugging Face Transformers 4.53.2 for model loading, compression, and inference. All evaluations are conducted in inference mode without gradient computation.

### 4.1 EVALUATION METRIC

We introduce several metrics to evaluate both the cross-dataset compressibility of various LLMs and the end-to-end performance of their compressed versions.

**Normalized Effective Rank**    We quantify the *data-dependent, per-layer* compressibility of the KV-cache using the NER (Roy & Vetterli, 2007), as introduced in Section 2.4. NER captures how evenly the singular values are distributed and yields a score in $[0, 1]$, with lower values indicating spectra dominated by a few large singular values and thus higher compressibility.

---

[1]Gemma-1.1: https://huggingface.co/google/gemma-1.1-7b-it

[2]glaiveai/glaive-function-calling-v2: https://huggingface.co/datasets/glaiveai/glaive-function-calling-v2

**Perplexity**  We evaluate the performance of compressed models using perplexity (PPL) (Bengio et al., 2003), the standard metric for language modeling. In practice, PPL is computed as the exponential of the empirical cross-entropy between the data distribution and the model distribution, which reduces to the Shannon entropy when the model perfectly matches the true distribution. Lower PPL indicates that the model assigns a higher probability to the observed data, while higher PPL reflects greater uncertainty or degraded predictive performance. Given retain ratio $k$ and $v$, the perplexity will be denoted as $\text{PPL}(k, v)$.

**Normalized Delta-Perplexity**  To directly measure the impact of KV-cache compression on model performance, we propose a quantitative metric, Normalized Delta-Perplexity (ND-PPL) for keys and values, denoted $\text{ND-PPL}_K$ and $\text{ND-PPL}_V$. Raw perplexity or absolute changes are not directly comparable across datasets, since their scale depends on the baseline. ND-PPL addresses this by normalizing pairwise perplexity differences across retained rank ratios by the corresponding baseline perplexity, and averaging over all candidate settings. This provides a dataset-agnostic measure of robustness under compression and establishes a direct link to the NER. The formal definition is as follows:

Let $\mathcal{K} = k_1, \ldots, k_m$ denote the set of retained rank ratios for the key matrices, where $k_i \in (0, 1]$; the definition for values $\mathcal{V} = v_1, \ldots, v_n$ is analogous. For a fixed value ratio $v \in \mathcal{V}$, we consider all pairs $(k_i, k_j) \in \mathcal{P}_K$, where $\mathcal{P}_K = (k, k') \in \mathcal{K} \times \mathcal{K} \mid k > k'$, and define the key-side metric $\text{ND-PPL}_K$:

$$\text{ND-PPL}_K = \frac{1}{|\mathcal{V}|} \sum_{v \in \mathcal{V}} \left[ \frac{1}{|\mathcal{P}_K|} \cdot \sum_{(k_i, k_j) \in \mathcal{P}_K} \left( \frac{\text{PPL}(k_j, v) - \text{PPL}(k_i, v)}{\text{PPL}(k_i, v)} \right) \right] \quad (5)$$

The definition of $\text{ND-PPL}_V$ is symmetric, obtained by fixing $k \in \mathcal{K}$ and averaging over pairs of $v_i, v_j \in \mathcal{V}$.

**GPT Score**  We assess whether compression preserves response quality using GPT-4o. For each test case, GPT-4o is provided with the input instruction together with the two responses from the original and compressed models. Under a fixed prompt, GPT-5 outputs a binary score: 1 if the two responses are judged roughly equal in quality, and 0 otherwise. Rough equivalence requires both answers to be reasonable and aligned with the instruction, tolerating stylistic differences but penalizing empty, irrelevant, or nonsensical outputs from the compressed model. The full system prompt used for evaluation is provided in Appendix B.

### 4.2 BENCHMARKING EXPERIMENT ON KV-CACHE COMPRESSIBILITY

#### 4.2.1 AVERAGE NER ACROSS MODELS AND DATASETS

Table 1 reports the mean NER of keys (NER-K) and values (NER-V), averaged over all attention layers across seven models on five English datasets and fifteen languages from VisR-Bench. Several trends emerge:

**Keys are consistently more compressible than values.**  Across all models and datasets, NER-K is substantially lower than NER-V, indicating that the key cache admits a more pronounced low-rank structure. This asymmetry highlights that compression techniques targeting keys may achieve higher savings with less impact on performance, whereas values tend to retain higher-rank structure and thus are less compressible.

**Cross-lingual variation outweighs cross-domain variation.**  For English datasets spanning different domains (e.g., Alpaca vs. FunctionCall), NER values remain relatively stable, suggesting that domain shifts do not strongly affect rank structure. In contrast, multilingual results reveal far greater variation, with languages such as Czech and German showing higher NER compared to Arabic and Finnish. This suggests that linguistic diversity, tokenization differences, and training data availability play a larger role than domain differences in determining compressibility.

**KV Capacity governs compressibility.**  We observe that the original KV dimension strongly affects model compressibility. Earlier models such as LLaMA-2-7B exhibit substantially lower NER compared to other models, likely due to larger key/value dimensions and weaker utilization

| Datasets | Qwen3-4B | | Qwen3-8B | | Gemma-2B | | Gemma-7B | | Mistral-7B | | Phi-3 | | LLaMA-2-7B | |
|---|---|---|---|---|---|---|---|---|---|---|---|---|---|---|
| | K | V | K | V | K | V | K | V | K | V | K | V | K | V |
| *Multi-domain Datasets* | | | | | | | | | | | | | | |
| Alpaca | 0.428 | 0.724 | 0.452 | 0.753 | 0.612 | 0.900 | 0.359 | 0.469 | 0.449 | 0.773 | 0.409 | 0.616 | 0.023 | 0.464 |
| MedAlpaca | 0.429 | 0.723 | 0.452 | 0.752 | 0.594 | 0.889 | 0.344 | 0.455 | 0.441 | 0.771 | 0.403 | 0.604 | 0.176 | 0.310 |
| CodeAlpaca | 0.421 | 0.708 | 0.443 | 0.737 | 0.589 | 0.869 | 0.321 | 0.429 | 0.420 | 0.733 | 0.380 | 0.571 | 0.028 | 0.337 |
| WizardCoder | 0.425 | 0.726 | 0.447 | 0.753 | 0.597 | 0.889 | 0.329 | 0.445 | 0.420 | 0.750 | 0.385 | 0.587 | 0.265 | 0.304 |
| FunctionCall | 0.432 | 0.731 | 0.451 | 0.756 | 0.608 | 0.900 | 0.342 | 0.458 | 0.432 | 0.762 | 0.397 | 0.604 | 0.135 | 0.449 |
| Average | 0.424 | 0.717 | 0.446 | 0.745 | 0.597 | 0.884 | 0.337 | 0.448 | 0.430 | 0.753 | 0.393 | 0.593 | 0.088 | 0.141 |
| *Multilingual Question in VisR-Bench Datasets* | | | | | | | | | | | | | | |
| Czech | 0.383 | 0.627 | 0.401 | 0.652 | 0.536 | 0.803 | 0.292 | 0.383 | 0.385 | 0.660 | 0.313 | 0.470 | 0.099 | 0.148 |
| German | 0.383 | 0.640 | 0.400 | 0.666 | 0.536 | 0.802 | 0.305 | 0.400 | 0.392 | 0.676 | 0.345 | 0.523 | 0.092 | 0.138 |
| Italian | 0.377 | 0.632 | 0.392 | 0.655 | 0.529 | 0.793 | 0.299 | 0.393 | 0.387 | 0.669 | 0.339 | 0.515 | 0.084 | 0.135 |
| Dutch | 0.376 | 0.628 | 0.395 | 0.657 | 0.534 | 0.802 | 0.299 | 0.394 | 0.385 | 0.664 | 0.331 | 0.499 | 0.116 | 0.108 |
| Croatian | 0.375 | 0.620 | 0.393 | 0.645 | 0.525 | 0.791 | 0.292 | 0.382 | 0.383 | 0.659 | 0.320 | 0.479 | 0.081 | 0.132 |
| French | 0.373 | 0.633 | 0.390 | 0.657 | 0.532 | 0.797 | 0.301 | 0.397 | 0.387 | 0.671 | 0.340 | 0.519 | 0.087 | 0.137 |
| Vietnamese | 0.373 | 0.625 | 0.392 | 0.653 | 0.542 | 0.814 | 0.291 | 0.384 | 0.367 | 0.630 | 0.305 | 0.442 | 0.071 | 0.119 |
| Swedish | 0.371 | 0.620 | 0.387 | 0.645 | 0.526 | 0.794 | 0.296 | 0.389 | 0.381 | 0.656 | 0.322 | 0.486 | 0.078 | 0.128 |
| Spanish | 0.367 | 0.629 | 0.384 | 0.654 | 0.523 | 0.790 | 0.299 | 0.396 | 0.381 | 0.665 | 0.334 | 0.513 | 0.064 | 0.095 |
| Slovenian | 0.366 | 0.603 | 0.383 | 0.628 | 0.518 | 0.785 | 0.281 | 0.369 | 0.372 | 0.642 | 0.306 | 0.458 | 0.075 | 0.121 |
| Portuguese | 0.362 | 0.614 | 0.378 | 0.639 | 0.521 | 0.785 | 0.284 | 0.375 | 0.379 | 0.656 | 0.326 | 0.498 | 0.083 | 0.133 |
| Japanese | 0.361 | 0.619 | 0.380 | 0.648 | 0.506 | 0.775 | 0.276 | 0.369 | 0.364 | 0.636 | 0.321 | 0.467 | 0.079 | 0.125 |
| Finnish | 0.360 | 0.597 | 0.378 | 0.625 | 0.502 | 0.769 | 0.280 | 0.369 | 0.353 | 0.616 | 0.305 | 0.455 | 0.073 | 0.118 |
| Arabic | 0.337 | 0.582 | 0.354 | 0.609 | 0.503 | 0.769 | 0.270 | 0.361 | 0.338 | 0.594 | 0.288 | 0.413 | 0.052 | 0.173 |
| Average | 0.387 | 0.653 | 0.407 | 0.679 | 0.548 | 0.822 | 0.306 | 0.405 | 0.395 | 0.684 | 0.345 | 0.520 | 0.083 | 0.134 |

Table 1: Average NER of keys and values across all layers of 7 models on multilingual split of VisR-bench covering 15 languages

of rank capacity during training. Within the Gemma family, Gemma-7B shows much lower NER than Gemma-2B, which can be explained by its 16× larger KV dimension (16 heads × 256 per head) compared to the 2B model (1 head × 256). The higher compressibility of Gemma-7B suggests that this expanded KV capacity is under-utilized, whereas the smaller 2B models may make more efficient use of their available dimensions. In contrast, Qwen3-4B and Qwen3-8B show only minor differences in NER because both adopt the same compact KV configuration (8 heads × 128 per head).

**Rank collapse in low-resource languages.** Certain languages with limited pretraining coverage (e.g., Arabic, Slovenian, Finnish) exhibit unusually low NER, especially in the value cache. This phenomenon may reflect under-trained token embeddings collapsing into low-dimensional subspaces. Beyond compressibility, such rank collapse may serve as a diagnostic signal for identifying under-represented languages in multilingual pretraining corpora.

### 4.2.2 LAYER-WISE COMPRESSIBILITY PATTERNS

Figure 2 presents the layer-wise NER of Qwen3-4B on the five datasets and on three selected language subsets of VisR-Bench, with additional plots for other models included in Appendix C.1. The results show that NER is not uniform across layers. Middle layers often exhibit higher NER, suggesting they make fuller use of their representational capacity, while early and late layers are typically more compressible.

This heterogeneity indicates that future KV-cache compression should be layer-aware: applying a uniform compression ratio risks overly degrading high-rank layers while missing opportunities for more aggressive reduction in lower-rank ones. Moreover, the relative positions of high- and low-rank layers are consistent across datasets, suggesting that compressibility is partly a structural property of the model rather than purely data-driven. At the same time, subtle differences between tasks and languages (e.g., Arabic vs. English subsets) highlight that dataset characteristics can modulate layer usage, pointing to potential for data-aware compression strategies.

### 4.3 PERFORMANCE IMPACT OF KV-CACHE COMPRESSION

We evaluate the performance degradation of compressed models using both perplexity (PPL) and GPT-score. Figure 3 presents PPL heatmaps of Qwen3-4B and LLaMA-2-7B on the Alpaca dataset

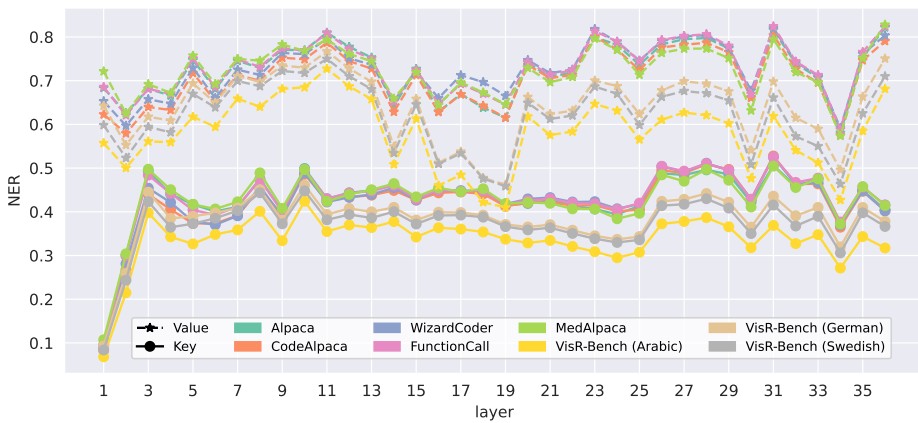

Figure 2: Layer-wise NER of key and value representations in Qwen3-4B, evaluated on 5 datasets and 3 languages from the VisR-Bench benchmark.

across a grid of KV-cache compression ratios, with additional results provided in Appendix C.2. We can see that LLaMA-2-7B remains relatively stable, showing only modest PPL increases even under aggressive compression, whereas Qwen3-4B is more sensitive, exhibiting substantial degradation. These results suggest that models with lower NER values are generally more compressible, as reflected by smaller changes in PPL.

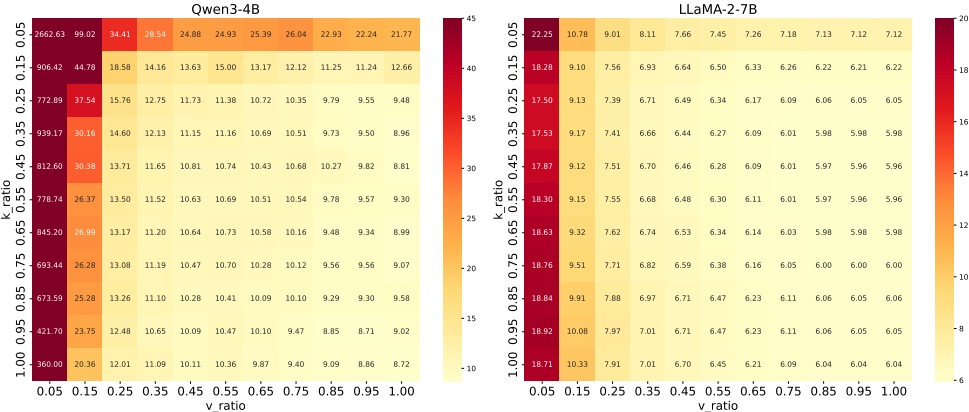

Figure 3: PPL heatmap of Qwen3-4B and LLaMA-2-7b on the Alpaca dataset.

In addition to PPL, we also report average GPT-score in Figure 4, which directly measures the quality difference between compressed and original model responses and provides a metric more closely aligned with user experience. Due to computational cost, GPT-scores are computed as averages over 100 instructions from the Alpaca dataset. Consistent with PPL trends, LLaMA-2-7B again proves more compressible than Qwen3-4B. Importantly, GPT-score further reveals a smoother and more continuous trajectory of performance degradation, even in regions where PPL remains relatively unchanged.

## 4.4 DATASET COMPRESSIBILITY COMPARISON

To quantitatively assess how well the NER reflects end-to-end robustness under compression, we employ the ND-PPL metrics. As reported in Figure 5, NER and ND-PPL are positively correlated, with Pearson $r = 0.88$ for values and $r = 0.64$ for keys. These results establish NER as a reliable predictor of performance sensitivity under compression. Moreover, the scatter plots reveal systematic dataset-level patterns. Multilingual datasets (e.g., Arabic, Portuguese, Finnish) cluster toward the bottom-left, with both low NER and low ND-PPL, indicating higher resilience to compression. In

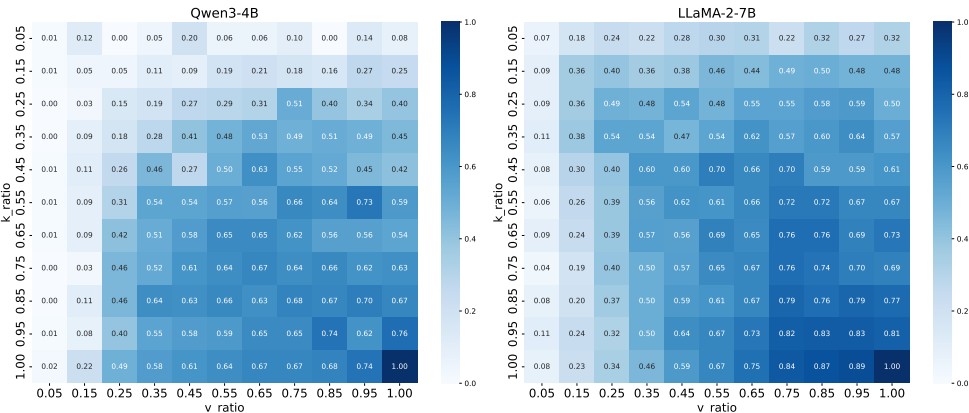

Figure 4: GPT score heatmap of Qwen3-4B and LLaMA-2-7b on the Alpaca dataset.

contrast, English-domain datasets such as Alpaca, MedAlpaca, and FunctionCall appear toward the upper-right, showing higher NER and greater sensitivity to compression. This separation suggests that KV-cache compressibility can serve as a diagnostic of data–model alignment: under-trained or poorly covered datasets tend to yield low NER, while well-represented domains exhibit higher NER and are more sensitive to aggressive truncation.

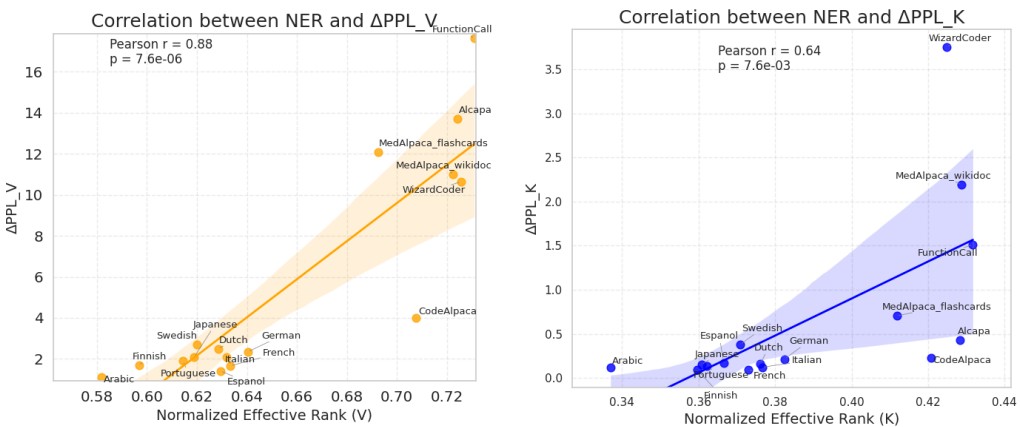

Figure 5: Correlation between dataset-level NER and ND-PPL computed by Qwen3-4B.

# 5 CONCLUSION

In this work, we introduced KV-CoRE, an SVD-based framework for dataset-level analysis of KV-cache compressibility in large language models. KV-CoRE directly decomposes cached key/value activations with low memory overhead, yielding globally optimal low-rank approximations and enabling systematic evaluation of rank utilization across layers and datasets.

Through extensive experiments across multiple model families, domains, and languages, we showed that NER serves as a lightweight and reliable indicator of compressibility, correlating closely with perplexity- and GPT-based performance under compression. We further introduced ND-PPL as an end-to-end robustness measure, establishing a clear empirical link between NER and model sensitivity to truncation.

Our analysis uncovers consistent patterns that tie compressibility to architectural design, training data, and language coverage. These findings position KV-CoRE as both a diagnostic tool for understanding representational efficiency and a benchmark for guiding the development of dynamic, data-aware KV-cache compression strategies and data-centric model improvements.

## 6 LANGUAGE MODEL USAGE STATEMENT

In preparing this manuscript, we used GPT-5 only for grammar checking and minor language polishing. The authors reviewed and edited all suggestions. All scientific content, analysis, and conclusions are entirely the work of the authors.

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

# Appendix

## A  PRELIMINARY

### A.1  MHA, MQA AND GQA

We first introduce the standard MHA and two of its variants—MQA and GQA. Given an input embedding vector, MHA project it into key and value vectors for each attention head, causing the KV cache size to scale linearly with the number of heads. In contrast, MQA and GQA reduce the KV cache size by grouping heads and sharing the same key and value vectors within each group.

Focusing on GQA, the most general technique among the three, we define $d_e$, $m_h$, $d_h$ and $m_g$ as the embedding dimension, number of heads, dimension per head and number of groups, respectively. Given an input embedding vector $\mathbf{x}_t \in \mathbb{R}^{1 \times d_e}$ corresponding to the t-th token, GQA divides the $m_h$ attention heads into $m_g$ groups. Formally, this grouping can be described by a helper function $g$, which maps from head indices $\{1, ..., m_h\}$ to group indices $\{1, ..., m_g\}$ as,

$$g(i) = \left\lceil i \bigg/ \frac{m_h}{m_g} \right\rceil, \quad \forall i \in \{1, ..., m_h\} \tag{A.1}$$

Then it projects $\mathbf{x}_t$ into query $\mathbf{q}_t \in \mathbb{R}^{1 \times m_h d_h}$, key $\mathbf{k}_t \in \mathbb{R}^{1 \times m_g d_h}$, and value vector $\mathbf{v}_t \in \mathbb{R}^{1 \times m_g d_h}$ as follows:

$$[\mathbf{q}_{t,1}, ..., \mathbf{q}_{t,m_h}] = \mathbf{q}_t = \mathbf{x}_t W^Q \tag{A.2}$$

$$[\mathbf{k}_{t,1}, ..., \mathbf{k}_{t,m_g}] = \mathbf{k}_t = \mathbf{x}_t W^K \tag{A.3}$$

$$[\mathbf{v}_{t,1}, ..., \mathbf{v}_{t,m_g}] = \mathbf{v}_t = \mathbf{x}_t W^V \tag{A.4}$$

where $\mathbf{q}_{t,i}, \forall i \in \{1, ..., m_h\}$ denote the query vector for each attention head, and $\mathbf{k}_{t,i}, \mathbf{v}_{t,i}, \forall i \in \{1, ..., m_g\}$ represent the key and value vector for each group. The matrices $W^Q \in \mathbb{R}^{d_e \times m_h d_h}$ and $W^K, W^V \in \mathbb{R}^{d_e \times m_g d_h}$ denote learnable model parameters. The attention of each head and the final projected output are computed as,

$$\mathbf{o}_{t,i} = \sum_{j=1}^{t} \text{Softmax}_j \left( \frac{\mathbf{q}_{t,i} \mathbf{k}_{j,g(i)}^T}{\sqrt{d_h}} \right) \mathbf{v}_{j,g(i)}, \quad \forall i \in \{1, ..., m_h\} \tag{A.5}$$

$$\mathbf{y}_t = [\mathbf{o}_{t,1}, ..., \mathbf{o}_{t,m_h}] W^O \tag{A.6}$$

where $\mathbf{o}_{t,i} \in \mathbb{R}^{1 \times d_h}$, $W^O \in \mathbb{R}^{m_h d_h \times d_e}$ and $\mathbf{y}_t \in \mathbb{R}^{1 \times d_e}$ denote the attention output for i-th head, the output projection matrix and the projected output respectively.

Note that when $m_g = m_h$, GQA reduces to standard MHA, and when $m_g = 1$, it specializes MQA.

### A.2  MLA

Unlike MHA and its variants, MLA projects the input embedding $\mathbf{x}_t \in \mathbb{R}^{d_e}$ of t-th token into two distinct spaces: a joint latent KV space and a decoupled key space designed to incorporate RoPE. Formally, this can be expressed as:

$$\mathbf{c}_t^{KV} \& = \mathbf{x}_t W^{DKV} \tag{A.7}$$

$$\mathbf{k}_t^R \& = \text{RoPE}(\mathbf{x}_t W^{KR}) \tag{A.8}$$

where $\mathbf{c}_t^{KV} \in \mathbb{R}^{1 \times d_c}$ and $\mathbf{k}_t^R \in \mathbb{R}^{1 \times d_R}$ denote the joint latent KV vector and the RoPE-encoded decoupled key vector, receptively. The projection matrices $W^{DKV} \in \mathbb{R}^{d_e \times d_c}$ and $W^{KR} \in \mathbb{R}^{d_e \times d_R}$ handle the corresponding down-projections. During attention calculation, $\mathbf{c}_t^{KV}$ is up-projected to get the key and value vectors, while the query vector is computed directly from the input embedding $\mathbf{x}_t$. These operations are described by following equations equation A.9 and equation A.10:

$$[\mathbf{k}_{t,1}^C, ..., \mathbf{k}_{t,n}^C] = \mathbf{k}_t^C = \mathbf{c}_t^{KV} W^{UK} \qquad \mathbf{c}_t^Q = \mathbf{x}_t W^{DQ}$$

$$[\mathbf{q}_{t,1}^C, ..., \mathbf{q}_{t,m_h}^C] = \mathbf{q}_t^C = \mathbf{c}_t^Q W^{UQ}$$

$$\mathbf{k}_{t,i} = [\mathbf{k}_{t,i}^C, \mathbf{k}_t^R] \qquad \text{(A.9)} \qquad [\mathbf{q}_{t,1}^R, ..., \mathbf{q}_{t,m_h}^R] = \mathbf{q}_t^R = \text{RoPE}(\mathbf{c}_t^Q W^{QR})$$

$$[\mathbf{v}_{t,1}^C, ..., \mathbf{v}_{t,m_h}^C] = \mathbf{v}_t^C = \mathbf{c}_t^{KV} W^{UV} \qquad \mathbf{q}_{t,i} = [\mathbf{q}_{t,i}^C, \mathbf{q}_{t,i}^R]$$

$$\text{(A.10)}$$

where $W^{UK}, W^{UV} \in \mathbb{R}^{d_c \times m_h d_h}$ are up-projection matrices for key and value vectors, respectively. The matrices $W^{DQ} \in \mathbb{R}^{d_e \times d_c'}$ and $W^{UQ} \in \mathbb{R}^{d_c' \times m_h d_h}$ serve as the down- and up-projection for queries, while $W^{QR} \in \mathbb{R}^{d_c' \times m_h d_h}$ is the up-projection matrix used to incorporate RoPE for the decoupled query vector. Note that both $\mathbf{k}_{t,i}$ and $\mathbf{q}_{t,i}$ are concatenations of their NoPE and RoPE components.

Using $\mathbf{q}_{t,i}$, $\mathbf{k}_{t,i}$ and $\mathbf{v}_{t,i}^C$, the attention of each head and the final projected output are computed as,

$$\mathbf{o}_{t,i} = \sum_{j=1}^{t} \text{Softmax}_j \left( \frac{\mathbf{q}_{t,i} \mathbf{k}_{j,i}^T}{\sqrt{d_h + d_R}} \right) \mathbf{v}_{j,i}^C, \quad \forall i \in \{1, ..., m_h\} \tag{A.11}$$

$$\mathbf{y}_t = [\mathbf{o}_{t,1}, ..., \mathbf{o}_{t,m_h}] W^O \tag{A.12}$$

A key merit of MLA lies in its caching efficiency during token generation: only $\mathbf{c}_t^{KV}$ and $\mathbf{k}_t^R$ need to be cached, resulting in a cache size of $d_c + d_R$. Since both $d_c << m_h d_h$ and $d_R << m_h d_h$, MLA reduces KV cache size significantly compared to MHA, which requires caching $\mathbf{k}_t \in \mathbb{R}^{1 \times m_h d_h}$ and $\mathbf{v}_t \in \mathbb{R}^{1 \times m_h d_h}$ for t-th token.

## B GPT EVALUATION SYSTEM PROMPT

Below is the system prompt used to prompt GPT-4o as a text generation quality score in our experiment:

---

**System Prompt 1.** *You are an automatic evaluator for LLM responses. Your job: compare two candidate answers (A = original model, B = compressed model) to the same user prompt and output a binary score.*

*—-*

*Scoring Rules*

- *Output 1 if A and B are roughly equal in quality (not necessarily the same wording or level of detail, but comparable usefulness for the task).*
- *Output 0 otherwise.*

*—-*

*"Roughly equal" means:*

- *Both A and B provide a reasonable, relevant answer to the user's prompt.*
- *They satisfy the intent to a similar degree, with no material difference in correctness, completeness, usefulness, or safety.*
- *Differences in style, verbosity, order, or minor details do not matter if they don't affect usefulness.*
- *If both are equally poor or equally failed (e.g., both empty, both nonsense, both refuse without reason, or both severely hallucinated to a similar extent), score 1.*
- *If one contains nonsense, large repetition, emptiness, or serious hallucinations while the other does not, score 0.*
- *Special rule: The COMPRESSED MODEL's answer must itself be a reasonable response to the user prompt.*
- *If the compressed model gives an empty output, nonsense, off-topic text, or fails to address the question, score 0, even if the original answer is also poor.*

*—-*

*Evaluation checklist (internal only):*

1. *Task fulfillment & correctness: Does each answer address the user's ask accurately?*
2. *Coherence & specificity: Is each answer clear, non-contradictory, minimally redundant?*
3. *Grounding & hallucinations: Any invented facts or unsupported claims?*
4. *Completeness at the needed granularity: Are the essentials present?*
5. *Harm & policy: Safety/compliance roughly comparable?*
6. *Degenerate behaviors: empty output, nonsense, repetition, prompt copy, or off-topic.*

*—-*

*Decision rules:*

- *Score 1 if both are reasonable answers and land in the same quality band (Excellent/Adequate/Poor/Fail), close enough that a reasonable user would find them similarly useful (or similarly useless).*
- *Score 0 if any material gap exists, or if the compressed model fails to provide a reasonable answer.*

*—-*

*Formatting:*

- *Return only a single JSON object with no extra text: "score": 0 or "score": 1*

*Example:*
*[INPUT PROMPT]*
*Explain why the sky is blue.*
*[ANSWER A: ORIGINAL MODEL]*
*"The sky looks blue due to Rayleigh scattering of sunlight in Earth's atmosphere, which scatters shorter wavelengths like blue more strongly."*
*[ANSWER B: COMPRESSED MODEL] ""*
*Expected output: "score": 0*

---

# C    ADDITIONAL EXPERIMENT RESULTS

## C.1    LAYER-WISE NER RESULTS

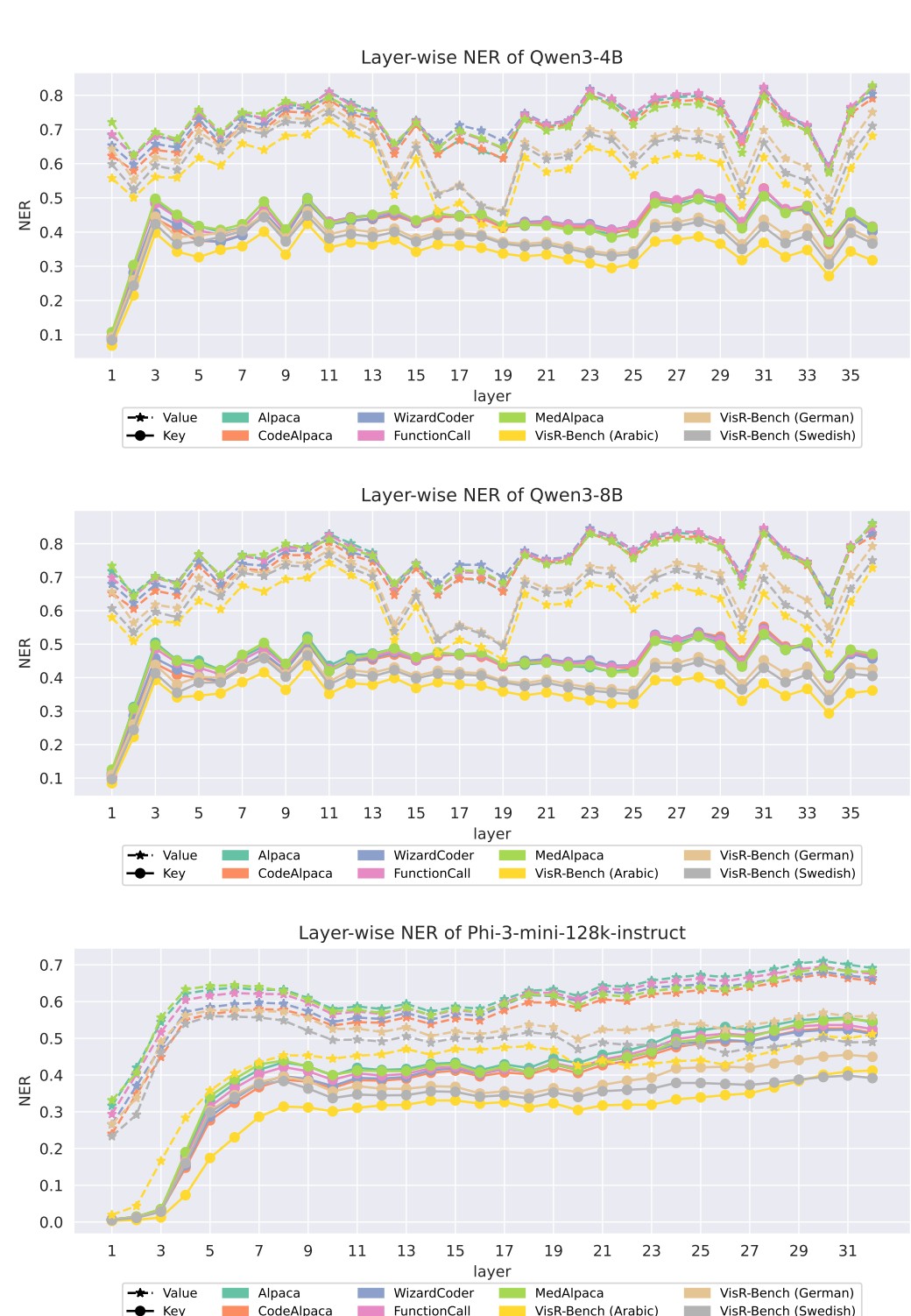

Figure C.1: Layer-wise NER of key and value representations in Qwen3-4B, Qwen3-8B, and Phi-3-mini evaluated on 5 datasets and 3 languages from the VisR-Bench benchmark.

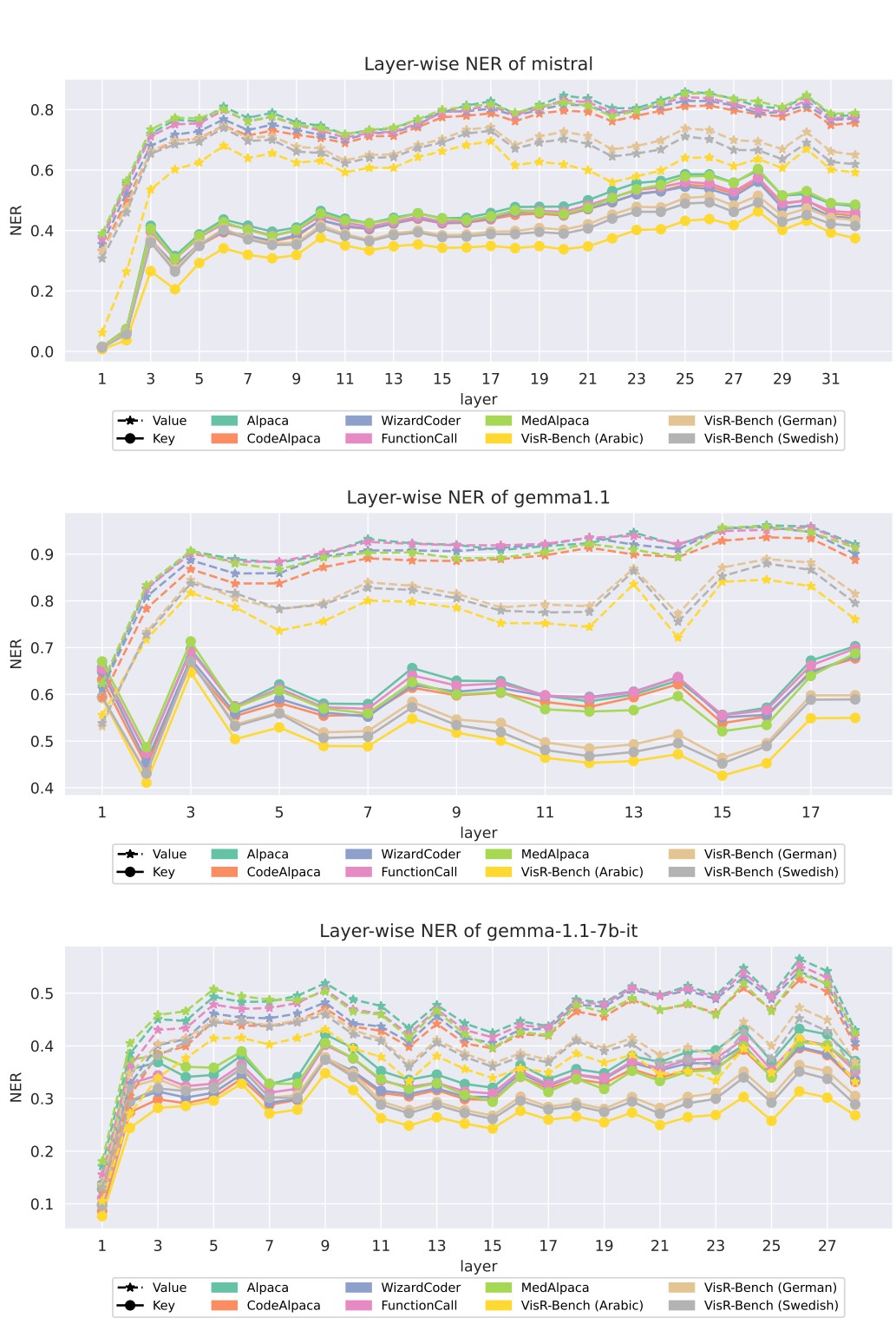

Figure C.2: Layer-wise NER of key and value representations in mistral, gemma1.1, and gemma-1.1-7b-it evaluated on 5 datasets and 3 languages from the VisR-Bench benchmark.

## C.2 PPL HEATMAP

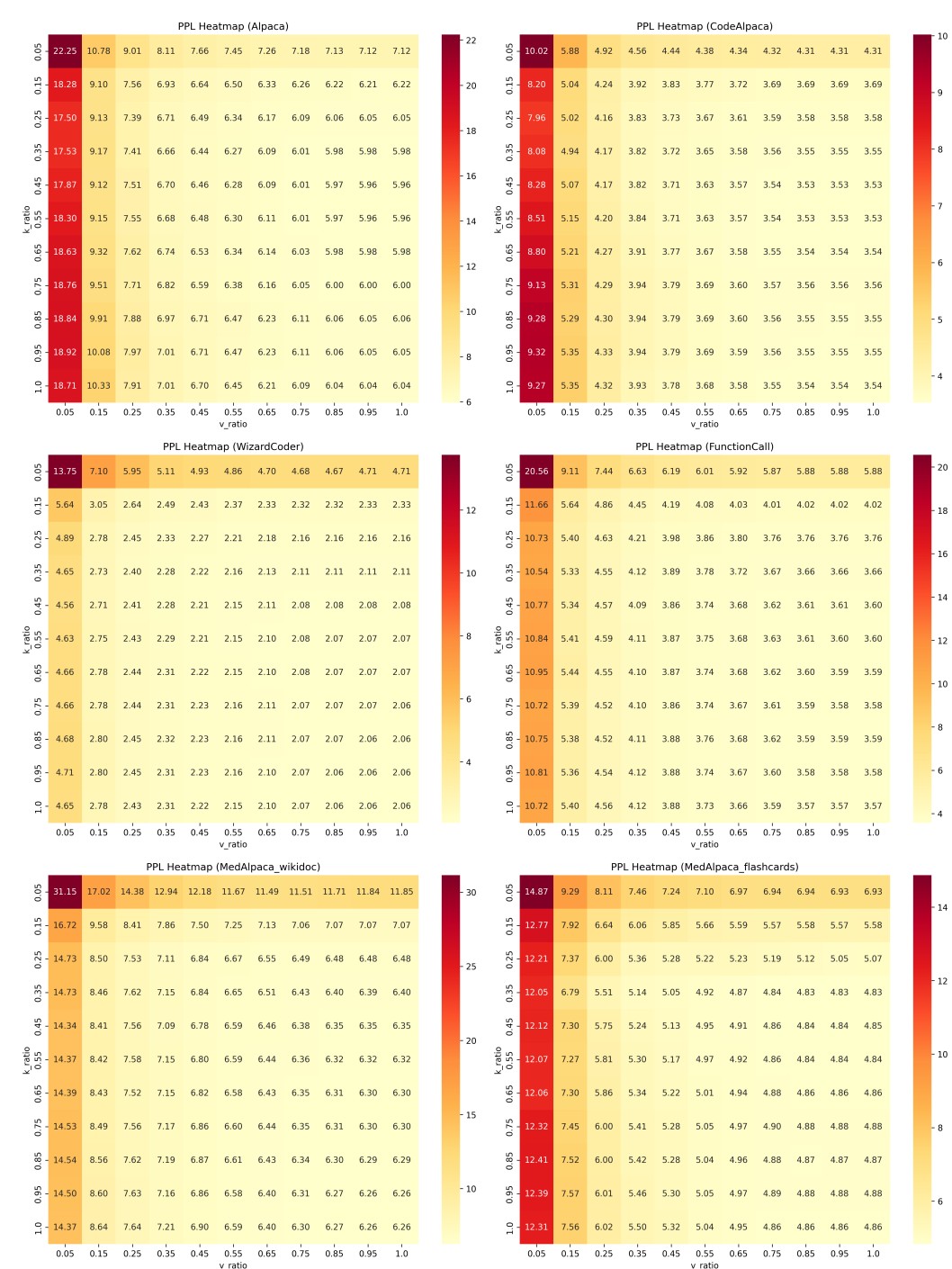

Figure C.3: PPL heatmap of LLaMA-2-7B on 6 datasets.

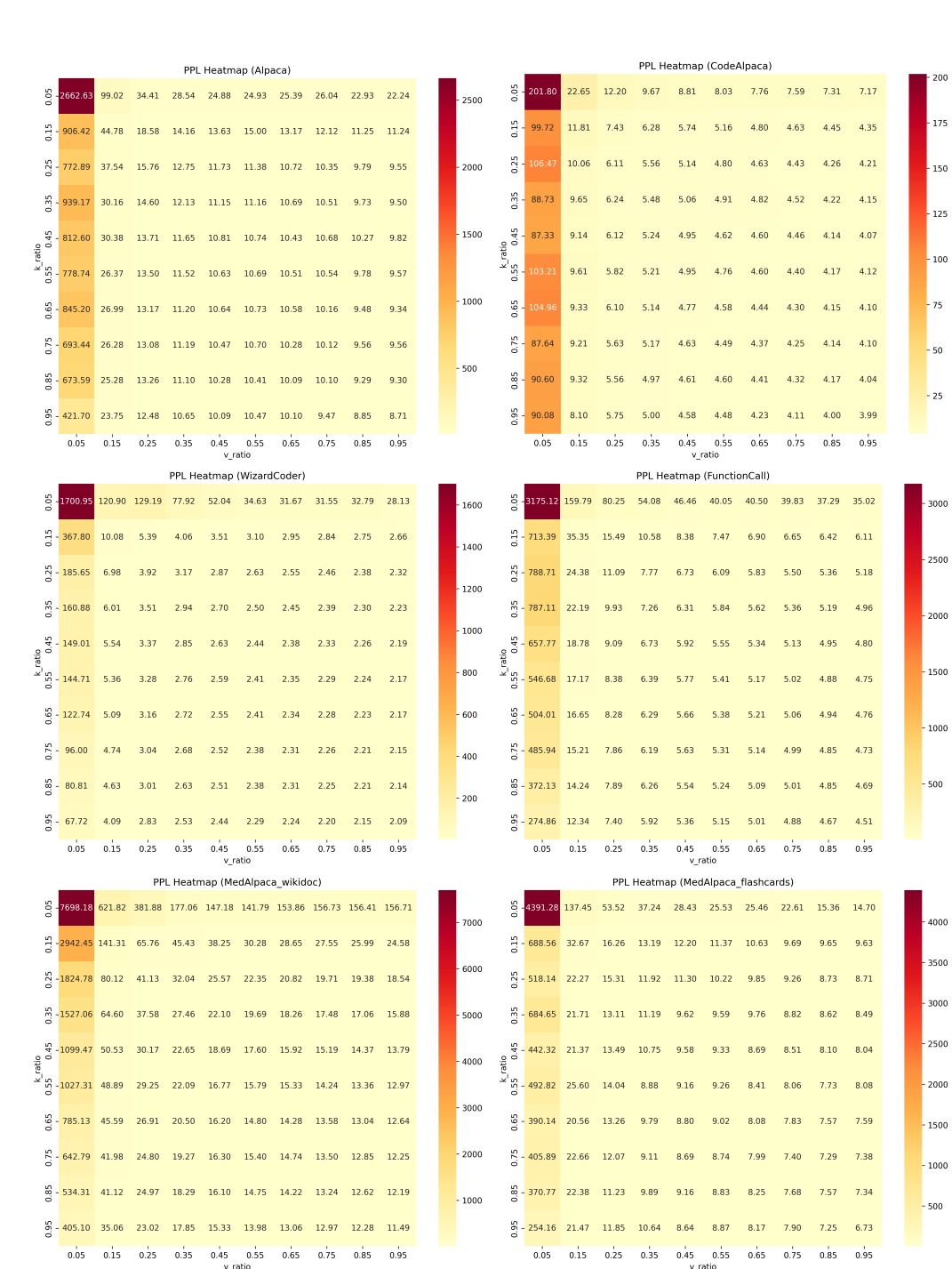

Figure C.4: PPL heatmap of Qwen3-4B on 6 datasets.

