# OpenReview forum: "KV-CoRE: Benchmarking Data-Dependent Low-Rank Compressibility of KV-Caches in LLMs"
_ICLR.cc/2026/Conference — ICLR 2026 Conference Withdrawn Submission_

### Official Review · Reviewer_3x9F · 2025-10-28

**Soundness:** 3
**Presentation:** 3
**Contribution:** 3
**Rating:** 6
**Confidence:** 3

**Summary:**

KV-CoRE introduces an SVD-based framework to measure how compressible LLM KV-caches are in a data-dependent way. It computes optimal low-rank approximations of key/value activations, defines a normalized effective rank (NER) metric that correlates with model performance under compression, and provides the first large-scale benchmark of cache compressibility across models, datasets, and languages. Results show that keys are more compressible than values, layer-wise and language-specific patterns exist, and NER predicts performance degradation under compression.

**Strengths:**

1. Provides a data-dependent, SVD-based framework for analyzing KV-cache compressibility with NER.
2. Demonstrates interesting results that there is strong correlation correlation between NER and real performance metrics (perplexity, GPT score).
3. Establishes the first large-scale benchmark for KV-cache compressibility across models and domains.

**Weaknesses:**

1. No improvement over previous methods but more like a case study.
2. Delta is not clear from SVD methods.

**Questions:**

1. What are some ways that people can further use this method for compression?
2. Which compression methods does the paper use in the reported results for compressibility? There are many papers on compressibility out there. Does the results hold for different types of kv compression?

---

### Official Review · Reviewer_3Lxo · 2025-10-30

**Soundness:** 3
**Presentation:** 3
**Contribution:** 3
**Rating:** 4
**Confidence:** 4

**Summary:**

The paper presents KV-CoRE, an SVD-based method for quantifying the data-dependent low-rank compressibility of KV-caches by computing the low-rank approximation under the Frobenius norm. They analysed multiple models and datasets spanning five English domains and sixteen languages to uncover systematic patterns linking compressibility to model architecture, training data, and language coverage.

**Strengths:**

* Interesting insight on using different datasets to incrementally improve the SVD calculation
* NER metric is introduced as a principled way to measure compressibility and also make it interpretable by tying it to known metrics such as perplexity and GPT score

**Weaknesses:**

* *Keys are consistently more compressible than values* I wish the paper provided some intuition on why this is the case beyond empirical evidence alone.

* The evaluation is constrained to a smaller family of models only (fewer than 10B parameters). Also, it is unclear whether techniques like grouped-query attention, which show greater effect in larger models, will affect the measurements done.

**Questions:**

* I would like to see the analysis over a larger suite of models (not just 4-8B parameters). Based on the text, the experimental setup the authors have is enough to carry out experiments on much larger models, so I am curious why the model set was so constrained to a small set of parameters.
* *Cross-lingual variation outweighs cross-domain variation* This insight seems a bit trivial, given that there are more similarities within a language across domains than similarities across languages. The paper can benefit from more study on why certain language families are more compressible than others, there might be more fundamental insights there. This can help strengthen the paper, as one of the core premises is studying the effect of the SVD based KV Compression across languages.

---

### Official Review · Reviewer_3bXN · 2025-10-30

**Soundness:** 2
**Presentation:** 2
**Contribution:** 2
**Rating:** 2
**Confidence:** 5

**Summary:**

This paper proposes an SVD-based method for quantifying the data-dependent low-rank compressibility of KV-caches, called KV-CoRE (KV-cache Compressibility by Rank Evaluation). The authors employ the Normalized Effective Rank (NER) as a metric of compressibility and show strong correlations between NER and performance degradation under compression, i.e., compressibility. More intuitively, NER provides a normalized, layer-wise indicator of how much a trunk of KV-cache can be low-rank approximated/compressed, i.e., how compressible the trunk of KV-cache is. NER is experimentally verified to be a successful compressibility predictor.

**Strengths:**

1. NER is experimentally verified to be a successful compressibility predictor, and can be used as an indicator of how much a trunk of KV-cache can be low-rank approximated/compressed.
2. Another major contribution of this work is the insights observed from experiments: KV-compressibility is layer- and data-dependent, and thus KV-cache compression should be layer-wise and data-aware.

**Weaknesses:**

1. Introducing or defining metric(s) for compressibility analysis is basically the core of many related (low-rank compression) methods. This paper mainly focuses on compressibility analysis, but does not proceed to really compress the models and demonstrate results of model compression, either. I would consider this work as yet-another-work on metric definition -- the novelty is not significant and the impact is not guaranteed.
2. Or, the authors need to reveal the superiority of the proposed NER by comparing it to other counterparts used/presented in existing representative papers. Toward this end, the authors may still need to compress a model based on NER and demonstrate NER-based model compression is better from the perspective of trade-offs between model compression and model performance.
3. Layer-wise analysis is demonstrated but the reviewer expected to see comparisons of layer-wise compression vs. uniform compression.

**Questions:**

My questions and suggestions are basically from "Weaknesses" as aforementioned.
1. From Weakness 1: Please address my concerns about the novelty and practical impact of this work.
2. From Weakness 2: Please demonstrate results of NER-based model compression, and compare the compression results to other existing counterparts.
3. From Weakness 3: Comparisons of layer-wise compression vs. uniform compression are preferred, and can further justify the observation that KV-cache compression should be layer-wise.

---

### Official Review · Reviewer_q3ax · 2025-11-01

**Soundness:** 2
**Presentation:** 2
**Contribution:** 2
**Rating:** 4
**Confidence:** 3

**Summary:**

- KV-CoRE introduces a data-dependent, SVD-based framework for analyzing and optimizing KV-cache compressibility in large language models.
- By incrementally computing the covariance of cached key/value activations, it efficiently derives per-layer singular value spectra without - storing all tokens, yielding provably optimal low-rank projections with minimal memory overhead.
- The paper further proposes the Normalized Effective Rank (NER) as a lightweight measure of compressibility, showing strong correlation with performance degradation under compression across models, domains, and languages.
- This establishes the first large-scale benchmark of KV-cache compressibility and provides practical insights for layer-aware, data-aware, and dynamic KV-cache compression in LLM inference.

**Strengths:**

- The paper shifts the focus from weight-based to data-dependent compressibility of KV-caches
- Its methodological quality is high—the approach is mathematically grounded in the Eckart–Young–Mirsky theorem, implements a computationally efficient incremental SVD algorithm, and provides clear optimality guarantees under the Frobenius norm.
- The work is also clear and well-structured, with precise notation, intuitive illustrations and rigorous yet accessible derivations.

**Weaknesses:**

- Although KV-CoRE claims to be computationally efficient, the paper provides no empirical measurements of runtime, memory consumption, or throughput improvements compared to baseline SVD or Cholesky-based methods (e.g., SVD-LLM, Wang et al., 2024).
- The method introduces choices such as batch size, covariance update frequency, and rank-selection strategy, yet their effects on accuracy and stability are unexplored. A brief ablation could clarify robustness and guide practical deployment.
- While the paper demonstrates that NER correlates with performance degradation, it stops short of proposing or testing adaptive compression schemes that use NER in real inference, missing an opportunity to demonstrate end-to-end benefits.
- Even though the paper’s goal is analytical rather than systems benchmarking, providing quantitative efficiency evidence would strengthen its empirical credibility. The authors claim that KV-CoRE performs incremental SVD efficiently (enabling large-scale, dataset-level evaluation “with low memory overhead” and “without sacrificing accuracy”), yet no runtime or memory results are shown. Likewise, since the framework is motivated by the KV-cache memory bottleneck, linking the proposed diagnostic (NER) to actual GPU savings or speedups would better demonstrate its practical value and substantiate the efficiency claims.

**Questions:**

Can the authors provide concrete measurements of runtime and memory savings during incremental SVD computation and compressed inference? For instance, how does KV-CoRE’s covariance-update approach compare to full SVD or Cholesky-based methods (e.g., SVD-LLM) in GPU memory use and wall-clock time per layer or per token?

Even though the goal of the paper is analytical benchmarking rather than systems benchmarking, quantitative efficiency evidence would strengthen the empirical credibility of two claims the authors make:

“Incremental SVD is efficient” — They repeatedly emphasize that KV-CoRE enables large-scale, dataset-level evaluation “with low memory overhead” and “without sacrificing accuracy.” Those are efficiency claims, so showing concrete runtime/memory comparisons (even a simple scaling table) would substantiate them.

“Compression saves resources” — The framework is motivated by the KV-cache memory bottleneck; even if NER is only a diagnostic metric, linking it to measurable GPU savings or speedups would clarify why this diagnostic matters in practice.

---

### Note · Authors · 2025-11-24

**Comment:**

We have decided to withdraw this submission from ICLR 2025 to further improve the work.
We thank the reviewers and the program committee for their time and constructive feedback.

**Withdrawal Confirmation:**

I have read and agree with the venue's withdrawal policy on behalf of myself and my co-authors.